# Conditional Permutation Invariant Flows

## Abstract

We present a novel, conditional generative probabilistic model of set-valued data with a tractable log density. This model is a continuous normalizing flow governed by permutation equivariant dynamics. These dynamics are driven by a learnable per-set-element term and pairwise interactions, both parametrized by deep neural networks. We illustrate the utility of this model via applications including (1) complex traffic scene generation conditioned on visually specified map information, and (2) object bounding box generation conditioned directly on images. We train our model by maximizing the expected likelihood of labeled conditional data under our flow, with the aid of a penalty that ensures the dynamics are smooth and hence efficiently solvable. Our method significantly outperforms non-permutation invariant baselines in terms of log likelihood and domain-specific metrics (offroad, collision, and combined infractions), yielding realistic samples that are difficult to distinguish from real data.

## 1 Introduction

Invariances built into neural network architectures can exploit symmetries to create more data efficient models. While these principles have long been known in discriminative modelling (Lecun et al., 1998; Cohen & Welling, 2015; 2016; Finzi et al., 2021), in particular permutation invariance has only recently become a topic of interest in generative models (Greff et al., 2019; Locatello et al., 2020). When learning a density that should be invariant to permutations we can either incorporate permutation invariance into the architecture of our deep generative model or we can factorially augment our observations and hope that the generative model architecture is sufficiently flexible to at least approximately learn a distribution that assigns the same mass to known equivalents. The former is vastly more data efficient but places restrictions on the kinds of architectures that can be utilized, which might lead one to worry about performance limitations. While the latter does allow unrestricted architectures it is often is so data-inefficient that, despite the advantage of fewer limitations, achieving good performance is extremely challenging, to the point of being impossible.

In this work we describe a new approach to permutation invariant *conditional* density estimation that, while architecturally restricted to achieve invariance, is demonstrably flexible enough to achieve high performance on a number of non-trivial density estimation tasks.

Permutation invariant distributions, where the likelihood of a collection of objects does not change if they are re-ordered, appear widely. The joint distribution of independent and identically distributed observations is permutation invariant, while in more complex examples the observations are no longer independent, but still exchangeable. Practical examples include the distribution of non-overlapping physical object locations in a scene, the set of potentially overlapping object bounding boxes given an image, and so forth (see Fig. 1). In all of these we know that the probability assigned to a set of such objects (i.e. locations, bounding boxes) should be invariant to the order of the objects in the joint distribution function argument list.

Recent work has addressed this problem by introducing equivariant normalizing flows (Köhler et al., 2020; Satorras et al., 2021; Biloš & Günnemann, 2021). Our work builds on theirs but differs in subtle but key ways that increase the flexibility of our models. More substantially this body of prior art focuses on *non-conditional* density estimation. The work of Satorras et al. (2021) does consider a form of implicit conditioning, where the flow is evaluated for different graph sizes. In this work we go beyond that by making the dynamics that constitute our flow dependent on a conditional input. To this end, we believe we are the first to develop *conditional* permutation invariant flows, that are *explicitly* dependent on external input.

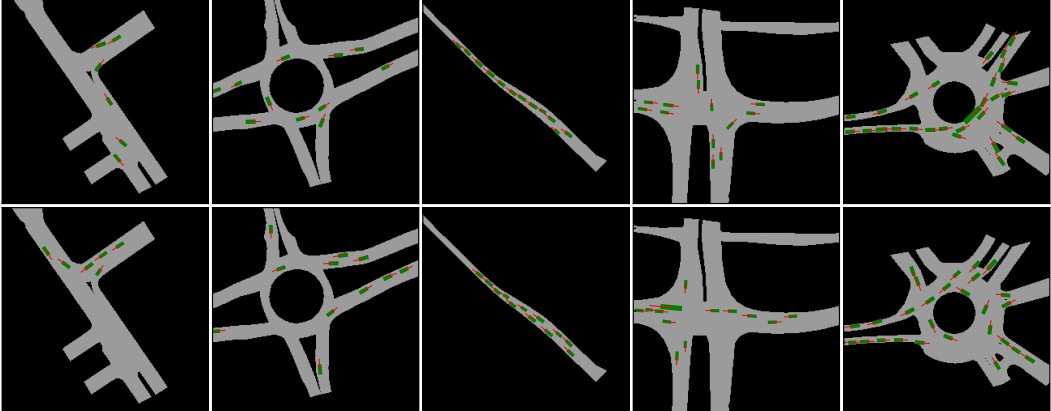

Figure 1: Realistic vehicle placement as a permutation invariant modeling problem. At every moment in time vehicles in the real world exhibit a characteristic spatial distribution of position, orientation, and size; notably vehicles (green rectangles) do not overlap, usually are correctly oriented (red lines indicate forward direction), and almost exclusively are conditionally distributed so as to be present only in driving lanes (shown in grey). The likelihood of each such arrangement does not depend on the ordering of the vehicles (permutation invariance). Each column shows a particular map with vehicle positions from real training data and from infraction free samples drawn from our permutation invariant flow conditioned on the map image. Note that because the image indicates lanes, not drivable area, the training data includes examples of vehicles that hang over into the black. We invite the reader to guess which image in each column is real and which is generated by our model. The answer appears in a footnote at the end of the paper.[1]

We demonstrate our conditional permutation invariant flow on two difficult conditional density estimation tasks: realistic traffic scene generation (Fig. 1) given a map and bounding box prediction given an image. In both the set of permutation invariant objects is a set of oriented bounding boxes with additional associated semantic information such as heading. We show that our method significantly outperforms baselines and meaningful ablations of our model.

## 1.1 BACKGROUND

### 1.1.1 NORMALIZING FLOWS

Normalizing Flows (Tabak & Vanden-Eijnden, 2010; Tabak & Turner, 2013; Rezende & Mohamed, 2015) are probability distributions that are constructed by combining a simple base distribution $p_{\mathbf{z}}(\mathbf{z})$ (e.g., a standard normal) and a differentiable transformation $T$ with differentiable inverse, that maps $\mathbf{z}$ to a variable $\mathbf{x}$

$$\mathbf{x} = T^{-1}(\mathbf{z}). \tag{1}$$

We can then express the density using the change of variables formula

$$p_{\mathbf{x}}(\mathbf{x}) = p_{\mathbf{z}}(T(\mathbf{x})) \left| \det \left. \frac{\partial T^{-1}(\mathbf{z})}{\partial \mathbf{z}} \right|_{\mathbf{z}=T(\mathbf{x})} \right|^{-1}, \tag{2}$$

where $p$ denotes the respective densities over variables $\mathbf{x}$ and $\mathbf{z}$ connected by transformation $T$ with inverse $T^{-1}$. The transformation $T$ can be parametrized and used to approximate some distribution over data $\mathbf{x} \sim \pi$ by maximizing the likelihood of this data under the approximate distribution using gradient descent. An important feature distinguishing normalizing flows from other models is that in addition to a method to generate samples they provide a tractable log density, enabling maximum likelihood training and outlier detection among others. This formulation, while powerful, has two noteworthy design challenges: the right hand side of Eq. (2) has to be efficiently evaluable and the aforementioned requirement that $T$ be invertible. The approach in the field generally is to define a chain of transformations $T_0 \circ \cdots \circ T_n$, each of which satisfy both conditions. In this manner, they can be comparatively simple, yet when joined together provide a flexible approximating family.

### 1.1.2 CONTINUOUS NORMALIZING FLOWS

Continuous normalizing flows were first introduced in Chen et al. (2018), and then further developed in Grathwohl et al. (2019). The concept is to use a continuous transformation of variables, described by dynamics function $\mathbf{v}$ parametrized by $t$ in the form of an ordinary differential equation (ODE)

$$\mathbf{x}(t_1) = \mathbf{x}(t_0) + \int_{t_0}^{t_1} \mathbf{v}_\theta(\mathbf{x}(t), t)dt. \tag{3}$$

We set $\mathbf{x}(t_0) = \mathbf{z}$ and $\mathbf{x}(t_1) = \mathbf{x}$, so that Eq. (3) provides our definition of $T^{-1}$ as defined in Eq. (1). Similarly, if we integrate backward in time from $t_1$ to $t_0$ we obtain $T$. The dynamics $\mathbf{v}_\theta(\mathbf{x(t)}, t)$ can be represented by a flexible function. As long as the dynamics function is uniformly Lipschitz continuous in $\mathbf{x}$ and uniformly continuous in $t$, the solution to the ODE is unique, and the transformation is invertible (Coddington & Levinson, 1955). In this case, we can write the probability density as another ODE (Grathwohl et al., 2019)

$$\frac{\mathrm{d} \log p_t(\mathbf{x}(t))}{\mathrm{d}t} = -\nabla_\mathbf{x} \cdot \mathbf{v}_\theta(\mathbf{x}(t), t). \tag{4}$$

The term on the right hand side is the *divergence* (not gradient) of the dynamics (sometimes equivalently written as the trace of the Jacobian, note that $\mathbf{v}_\theta$ is vector valued in Eq. (4)). Integrating this ODE from the probability density at $t_0$ gives the density at $t_1$

$$\log p_{t_1}(\mathbf{x}(t_1)) = \log p_{t_0}(\mathbf{x}(t_0)) - \int_{t_0}^{t_1} \nabla_\mathbf{x} \cdot \mathbf{v}_\theta(\mathbf{x}(t), t)dt. \tag{5}$$

Eq. (5) is the equivalent of Eq. (2) for continuous normalizing flows. Together with a suitable base distribution (e.g. a standard normal), the above transformation constitutes a distribution with a tractable likelihood and generative mechanism, which we will exploit to construct our flows.

### 1.1.3 INVARIANCE AND EQUIVARIANCE

We seek to construct distributions that have a permutation invariant density via permutation equivariant transformations. We state here the definition of permutation invariance and equivariance we adopt.

**Definition 1.** Let $\mathbf{x} = (\mathbf{x}_1 \dots \mathbf{x}_N)$ where each $\mathbf{x}_n \in \mathbb{R}^D$, and let permutations $\sigma$ act on $\mathbf{x}$ via

$$\sigma\mathbf{x} = (\mathbf{x}_{\sigma_1} \dots \mathbf{x}_{\sigma_N}). \tag{6}$$

A function $G : \mathbb{R}^{N \times D} \to \mathbb{R}$ is permutation *invariant* if for any permutation $\sigma$,

$$\forall \mathbf{x} \in \mathbb{R}^{N \times D}, \quad G(\sigma\mathbf{x}) = G(\mathbf{x}). \tag{7}$$

A function $F : \mathbb{R}^{N \times D} \to \mathbb{R}^{N \times D}$ is permutation *equivariant* if for any permutation $\sigma$,

$$\forall \mathbf{x} \in \mathbb{R}^{N \times D}, \quad F(\sigma\mathbf{x}) = \sigma F(\mathbf{x}). \tag{8}$$

## 1.2 RELATED WORK

Permutation invariant models have been studied in the literature for some time. Examples include models of sets (Zaheer et al., 2017; Lee et al., 2019), and graphs (Duvenaud et al., 2015; Kipf & Welling, 2017; Kipf et al., 2018). Recently, also generative models for sets have made an appearance, (Zhang et al., 2019; Burgess et al., 2019; Greff et al., 2019; Locatello et al., 2020; Zhang et al., 2020). Our conditional permutation invariant flows belong to the larger class of generative models, such as variational autoencoders (Kingma & Welling, 2014), generative adversarial networks (Goodfellow et al., 2014), and normalizing flows (Rezende & Mohamed, 2015). Among these, normalizing flows are the only class of models that enables likelihood evaluation.

Other work belonging to the generative category is "Equivariant Hamiltonian flows" (Rezende et al., 2019), which relates to our work since it models interactions elements of a set using Hamiltonian dynamics. The choice of these dynamics allows the use of a symplectic integrator, and the transformation is volume conserving, eliminating the need to integrate a divergence term. However, this requires the introduction of a set of momentum variables that preclude the exact calculation of a density. In Liu et al. (2019), the authors present a flow for graphs with tractable density. Although

the target domain is similar, their flow differs from ours as it is based on the mechanism developed in RealNVP (Dinh et al., 2017), whereas our flow uses continuous normalizing flows.

Our work is strongly related to, and draws inspiration from recent work that uses continuous normalizing flows with permutation invariant dynamics (Köhler et al., 2020; Satorras et al., 2021; Biloš & Günnemann, 2021). However Köhler et al. (2020) and Satorras et al. (2021) focus also on rotation and translation invariance, in order to model molecular graphs. Our work is also related to PointFlow, a continuous normalizing flow for point clouds (Li et al., 2021). We focus on sets like in Biloš & Günnemann (2021) and Li et al. (2021), however these studies focus on reducing evaluation cost for large set sizes. Our dynamics function on the other hand focuses on strong interactions between set elements, more akin to Satorras et al. (2021) and Köhler et al. (2020). Importantly, none of this previous work considers the problem of *conditioning* on external inputs and learning a distribution that is able to deal with a varying conditional input distribution.

## 2 CONDITIONAL PERMUTATION INVARIANT FLOWS

### 2.1 PERMUTATION INVARIANT FLOWS

In this work, we will construct normalizing flows that are characterized by a permutation equivariant transformation $T(\sigma\mathbf{x}) = \sigma T(\mathbf{x})$; we will demonstrate these flows produce a permutation invariant density $p(\mathbf{x}) = p(\sigma\mathbf{x})$. We construct our permutation invariant flows using a dynamics function that is based on a global force term and pairwise interaction terms

$$\mathbf{v}_{\theta,i}(\mathbf{x}) = \sum_{j, j \neq i} f_\theta(\mathbf{x}_i, \mathbf{x}_j) + g_\theta(\mathbf{x}_i). \tag{9}$$

Here, $\mathbf{v}_{\theta,i}(\mathbf{x}) \in \mathbb{R}^D$ denotes the $i^{\text{th}}$ element of $\mathbf{v}_\theta$, $\mathbf{x} \in \mathbb{R}^{N \times D}$, $g_\theta : \mathbb{R}^D \to \mathbb{R}^D$, and $f_\theta : \mathbb{R}^{2D} \to \mathbb{R}^D$. This construction can be interpreted as objects $\mathbf{x}_i$ moving in a global potential with corresponding force field $g_\theta(\mathbf{x}_i)$, and interacting with other objects through the pairwise interaction $f_\theta(\mathbf{x}_i, \mathbf{x}_j)$. We proceed to construct a continuous normalizing flow using the function in Eq. (9) as the dynamics. If we use a permutation invariant base distribution $p(\mathbf{x}(t_0)) = p(\mathbf{z})$ we obtain the following:

**Theorem 1.** *If the transformation $\mathbf{z} = T(\mathbf{x})$ defined in Eq. (3) has dynamics $\mathbf{v}_\theta(\mathbf{x})$ defined in Eq. (9), then $T$ is permutation equivariant. If in addition $p(\mathbf{z})$ is permutation invariant, then the density $p(\mathbf{x})$ is permutation invariant.*

The proof of Theorem 1 is given in Appendix A.1. We note that this theorem can be seen as a special case of theorems presented in previous work (Köhler et al., 2020; Zaheer et al., 2017), but add it here for a complete exposition. The dynamical system this theorem represents is similar to an interacting set of particles in a global potential. The dynamics as presented in Eq. (9) are independent of time; in a few cases, however, we have found it useful to make the dynamics time-dependent, i.e. $\mathbf{v}_\theta(\mathbf{x}, t)$, by passing time to both $g_\theta$ and $f_\theta$ as an input. A complete overview of when time dependence is used is given in the supplementary information Appendix A.2.3. In practice we represent $f_\theta$ and $g_\theta$ by neural networks, which satisfy the criterion of uniform Lipschitz continuity if activation functions are chosen appropriately, guaranteeing invertibility. Implementation details can be found in Appendix A.2.1.

### 2.2 DIVERGENCE

Given the dynamics in Eq. (9), we compute the density at time $t$ in Eq. (4) using the divergence

$$\nabla_\mathbf{x} \cdot \mathbf{v}_\theta(\mathbf{x}) = \sum_{i,j, j \neq i} \nabla_{\mathbf{x}_i} \cdot f_\theta(\mathbf{x}_i, \mathbf{x}_j) + \sum_i \nabla_{\mathbf{x}_i} \cdot g_\theta(\mathbf{x}_i). \tag{10}$$

A naïve computation of the divergence in Eq. (4) using automatic differentiation is expensive, as computing the Jacobian requires $ND$ evaluations, one for each of the $ND$ terms in $\mathbf{v}_\theta$ (Chen et al., 2018; Grathwohl et al., 2019). Since the cost of evaluating Eq. (9) is quadratic in $N$, this would result in an asymptotic computational cost of $N^3 D^2$ for the forward and backward pass. Earlier work has suggested the use of the Hutchinson's trace estimator (Chen et al., 2018) for the divergence, which reduces the cost of the divergence to that of $\mathbf{v}$, but suffers from high variance (Chen & Duvenaud,

2019). Instead we opt to re-express the divergence of $\mathbf{v}_\theta$ in terms of derivatives of $f(\mathbf{x}_i, \mathbf{x_j})$ and $g(\mathbf{x}_i)$, resulting in Eq. (10). The form in Eq. (10) is quadratic in $N$, and therefore same cost in $N$ as the evaluation of $\mathbf{v}_\theta$ itself, both in the forward and backward pass.

## 2.3 REGULARIZATION

Continuous normalizing flows have no inherent mechanism that penalizes very complex dynamics. While in theory there is no reason to prefer simple dynamics, in practice, the numerical integration of complex dynamics can result in long computation when using an adaptive scheme. This effect has been previously observed in the literature, and suggestions to regularize the dynamics have been proposed in previous work (Finlay et al., 2020; Kelly et al., 2020). While the work of Kelly et al. (2020) is more comprehensive, we find that an adaptation of the solution proposed in Finlay et al. (2020) works well for our purposes. The proposed solution in Finlay et al. (2020) is to add a term proportional to the squared Frobenius norm of the Jacobian, and the $\ell^2$-norm of the dynamics. We use the $\ell^2$-norm for the dynamics; however, since we do not estimate the full Jacobian we calculate

$$\ell_{div}^2 = \sum_{i \neq j, d, d'} \left( \frac{\partial f_d(x_{id}, x_{jd})}{\partial x_{id'}} \right)^2 + \sum_{i, d, d'} \left( \frac{\partial g_d(x_{id})}{\partial x_{id'}} \right)^2. \tag{11}$$

We find that this penalty significantly reduces the number of evaluations of our trained flows. We visualize the effect this penalty has on the dynamics in some examples in Appendix A.3.4.

## 2.4 CONDITIONAL PERMUTATION INVARIANT FLOWS

When performing amortized inference, in which a family of posterior distributions is learned, a requirement is a flexible variational family that can be made to depend (i.e. *conditioned*) on external input. An example would be to produce a valid distribution of a set of bounding box locations and sizes $\mathbf{x}$ for objects in an image $\mathbf{y}$ selected from a distribution of images. We will denote the conditioning input as $\mathbf{y}$, coming from some data distribution $\pi(\mathbf{y})$. To model such cases we construct a dynamics function that depends on $\mathbf{y}$ by modifying the pair forces, and the global force to $f_\theta(\mathbf{x}_i, \mathbf{x}_j, \mathbf{y})$ and $g_\theta(\mathbf{x}_i, \mathbf{y})$. The dynamics then become:

$$\mathbf{v}_{\theta,i}(\mathbf{x}, \mathbf{y}) = \sum_{j \setminus i} f_\theta(\mathbf{x}_i, \mathbf{x}_j, \mathbf{y}) + g_\theta(\mathbf{x}_i, \mathbf{y}). \tag{12}$$

Note that here too, the dynamics can be made time dependent by passing time $t$ as an argument.

We will train our flows by minimizing the Kullback-Leibler divergence to the joint distribution $\pi(\mathbf{x}, \mathbf{y}) = \pi(\mathbf{x}|\mathbf{y})\pi(\mathbf{y})$ over data $\mathbf{x}$ and condition $\mathbf{y}$:

$$\arg\min_\theta \mathrm{D_{KL}}\left( \pi(\mathbf{x}, \mathbf{y}) || p_\theta(\mathbf{x}|\mathbf{y})\pi(\mathbf{y}) \right) = \arg\min_\theta \mathbb{E}_{\mathbf{y} \sim \pi(\mathbf{y})} \mathrm{D_{KL}}\left( \pi(\mathbf{x}|\mathbf{y}) || p_\theta(\mathbf{x}|\mathbf{y}) \right). \tag{13}$$

In other words, we optimize our flow to match the distribution of $\mathbf{x}$ in *expectation* over $\pi(\mathbf{y})$.

## 3 EXPERIMENTS

### 3.1 SYNTHETIC EXAMPLES

We start our experiments with two pedagogical examples that demonstrate the capabilities and mechanisms of our flows. The first example task is to model the spatial distribution of five *non-overlapping* squares of width $w = 1$, that furthermore do not overlap with the prohibited regions shown in blue. This example is representative of placing assets into a physically realizable configuration in accordance with constraints imposed by an environment. We fit our conditional flow to a dataset generated by first sampling a prohibited region—three boxes of width $w = 1.5$ from a standard normal prior—and then sampling box locations independently from a standard normal prior, with rejection for overlap with previous boxes or the prohibited region, until a total of five boxes are sampled. The prohibited region is input to our conditional flow as an image tensor. Since the dataset was generated via rejection sampling, we can compare our sample efficiency against it. The conditional flow provides a substantial sampling efficiency improvement (77% valid) over rejection sampling (0.02% valid), in addition to a tractable density.

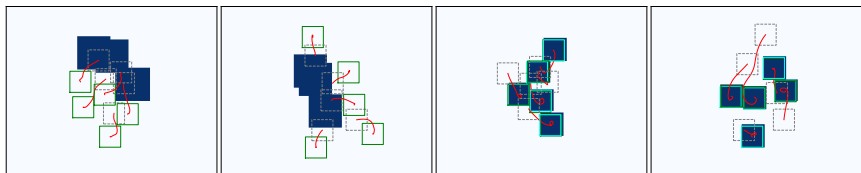

Figure 2: Two pedagogical permutation invariant modeling tasks. The left two panels illustrate the first task; *conditionally* modeling non-overlapping squares (green), which also do not overlap with the blue boxes whose arrangement varies between datapoints. The right two panels illustrate the second task; modeling boxes that are conditionally distributed so as to bound the underlying blue boxes. Samples from the base $p(\mathbf{z})$ and final distribution $p_\theta(\mathbf{x}|\mathbf{y})$ are plotted in dashed grey and green lines respectively. The conditional input is plotted as a blue on white image. Red lines indicate the trajectories the objects follow by integrating the dynamics function $\mathbf{v}(\mathbf{x}(t), \mathbf{y})$.

The second example task is bounding box prediction, or conditionally generating object bounding boxes $\mathbf{x}$ directly from an image $\mathbf{y}$. Here the objects are monochrome blue squares. Data is generated in a similar manner as in the first experiment: squares are sampled indepenently from a standard normal prior, and rejected if they overlap. The conditional input is an image of the generated boxes. Sampled bounding boxes from our trained flow achieve an average intersection over union (IOU) of 0.85 with the ground truth bounding boxes.

We display representative samples and their trajectories through time in Fig. 2 for both experiments. In the left two panels it can be seen that initial samples are transported around the space to avoid one another; in the right two, the boxes coordinate through the pairwise interactions to each surround exactly one of the objects in the scene. Further details for these experiments appear in Appendix A.3.

## 3.2 TRAFFIC SCENES

Modeling and being able to sample realistic traffic scenes is an essential task related to autonomous driving simulation and control. Referring back to Fig. 1, the problem—similar to the first pedagogical task above—is one of modeling the physical configuration a collection of agents conditioned on a representation of the environment. Until recently, the predominant methods for generating realistic vehicle configurations were rule-based (Yang & Koutsopoulos, 1996; Lopez et al., 2018). Rule-based systems can be tailored to have desirable properties such as avoiding occurences of offroad and vehicle overlap, but they produce vehicle arrangements that are distributionally dissimilar to real data.

Recent work addressing this problem uses non-rule-based autoregressive model (Tan et al., 2021) that enables sequential generation of vehicle and agent positions conditioned on a visual representation of a map. While this model-based approach closes the gap between simulation and reality, modeling sets autoregressively introduces a factorial data augmentation requirement, as there is no intrinsic ordering of actors. The authors of Tan et al. (2021) avoid this by imposing an arbitrary order, sampling agents from left to right. Our experiments indicate that, at least for this specific task, directly addressing permutation invariance is preferred, and avoids the need to arbitrarily fix the order of elements.

To test the performance of our flows on this task, we train them to generate a scene of cars in the INTERACTION dataset (Zhan et al., 2019), conditioned on a rendered image of the drivable area $\mathbf{y}$. The properties that our flows predict are two-dimensional position, size, aspect ratio and orientation for each of $N$ agents, i.e. $\mathbf{x} \in \mathbb{R}^{N \times 5}$. An advantage of the formulation of the dynamics in Eq. (12) is that they can be applied with the same $f$ and $g$ regardless of the number of agents $N$. We make use of this property, and train a single model on a varying amount of agents $N$. At test time, generating $N$ agents is accomplished simply by initializing the flow appropriately.

We train our flows until the likelihood of the data stops increasing, or the likelihood of a held out validation sets starts decreasing, whichever comes fist. Examples of the data we train on, and representative samples from our trained flow are shown in Fig. 1.

Table 1: Results for scene generation and bounding box prediction.

(a) Quantitative results for traffic scene generation. NLL indicates negative log-likelihood in nats, while the other metrics indicate the fraction of samples exhibiting offroad, collision, or either (lower is better).

| Method | NLL | Offroad | Collision | Infraction |
|---|---|---|---|---|
| Gaussian | $46.3 \pm 0.0$ | $0.99 \pm 0.00$ | $0.27 \pm 0.00$ | $0.99 \pm 0.00$ |
| RealNVP | $30.4 \pm 1.4$ | $0.96 \pm 0.01$ | $0.26 \pm 0.00$ | $0.98 \pm 0.00$ |
| CNF | $20.8 \pm 0.9$ | $0.86 \pm 0.01$ | $0.33 \pm 0.00$ | $0.92 \pm 0.01$ |
| Autoregr. | $11.0 \pm 1.7$ | $0.72 \pm 0.02$ | $0.14 \pm 0.01$ | $0.76 \pm 0.02$ |
| PIF Single | $7.3 \pm 0.2$ | $\mathbf{0.09 \pm 0.01}$ | $0.57 \pm 0.00$ | $0.61 \pm 0.01$ |
| PIF Pair | $\mathbf{6.3 \pm 0.3}$ | $0.17 \pm 0.01$ | $0.18 \pm 0.02$ | $0.32 \pm 0.02$ |
| PIF Pair MF | $7.0 \pm 0.1$ | $0.11 \pm 0.02$ | $0.56 \pm 0.01$ | $0.61 \pm 0.00$ |
| PIF (ours) | $\mathbf{6.5 \pm 0.3}$ | $0.12 \pm 0.01$ | $0.09 \pm 0.01$ | $\mathbf{0.20 \pm 0.01}$ |
| Cond. single | $7.1 \pm 0.3$ | $0.11 \pm 0.01$ | $0.12 \pm 0.00$ | $0.22 \pm 0.01$ |
| Cond. pair | $7.2 \pm 0.1$ | $0.18 \pm 0.02$ | $0.17 \pm 0.03$ | $0.32 \pm 0.03$ |
| Cond. base | $16.5 \pm 2.2$ | $0.91 \pm 0.01$ | $\mathbf{0.05 \pm 0.01}$ | $0.92 \pm 0.01$ |

(b) Quantitative results for bounding box prediction. IOU refers to the intersection over union of object area covered by the samples (higher is better).

| Data | IOU |
|---|---|
| 3 | 0.732 |
| 6-3 | 0.759 |
| 6-4 | 0.711 |
| 6-5 | 0.644 |
| 6-6 | 0.596 |
| 6 | 0.679 |

### 3.2.1 BASELINES

We compare our conditional flows against several baselines. Three are non-permutation invariant flows: a unimodal Gaussian model, a RealNVP based model (Dinh et al., 2017), and a "vanilla" continuous normalizing flow ("CNF"). We also implement an autoregressive model consisting of a convolutional neural net paired with a recurrent neural network and a 10 component Gaussian mixture for every prediction component, and adopt the canonical ordering discussed in Tan et al. (2021). We also test two ablations of our model: one where the dynamics are restricted only to single particle terms $g_\theta$ ("PIF Single"), and one the dynamics only include the pair term $f_\theta$ ("PIF Pair"). The flow where the dynamics are restricted to single particle terms only corresponds to a conditional version of PointFlow (Li et al., 2021). We also compare against a conditional version of the model presented in Biloš & Günnemann (2021) ("PF Pair MF"), in which interactions are addressed, but only in terms of a mean field (being cheaper to calculate). Both these methods result in a significantly worse collision and resulting infraction rate.

To compare to non-permutation invariant methods, we have to fix the number of agents, as these architectures cannot straightforwardly be provisioned to generate and score sets that differ in cardinality. We exclude all data with less than seven agents, and cases with more than seven agents are pruned to retain only the agents closest to the center. The cardinality of seven was chosen to retain as much data as possible while not making each individual scene too small. Furthermore, we restricted the INTERACTION dataset to the roundabout scenes in order to better match the seven agent target while still maintaining a semantically similar set of possible $\mathbf{y}$. We compare the negative log likelihood (NLL) of the various models on a held-out test dataset. For these traffic scenarios there are two other useful metrics we can compare: the fraction of offroad (i.e. an agent is on the undrivable area), collision (i.e., an agent overlaps with another agent) and total infractions (offroad or collision) the model makes. We note that these metrics are sensitive indicators of model fitness, in the sense that the training data contains no offroad or colliding data examples. Samples that exhibit these infractions are evidence of model error, and additionally inform whether modeling mistakes are made globally (i.e. offroad) or through interactions (i.e. collisions). We report our results in Table 1.

Comparing the likelihood and infraction metrics demonstrates the clear advantage of using a permutation invariant model. The non-permutation invariant version of our flow does not converge to a competitive likelihood, and struggles to generate infraction-free examples. While the canonically ordered autoregressive model is much more capable than the non-permutation invariant flows, it still underperforms compared to ours. The two ablations of our model provide an insightful result: the function of $g_\theta$ is to define the collective behavior of the agents (all of them need to stay on the road, independently of one another), whereas $f_\theta$ provides the necessary interaction between them (agents should not collide with one-another). These functions are evident from the respective infraction metrics. Furthermore, there appears to be a certain amount of competition between the pair and single

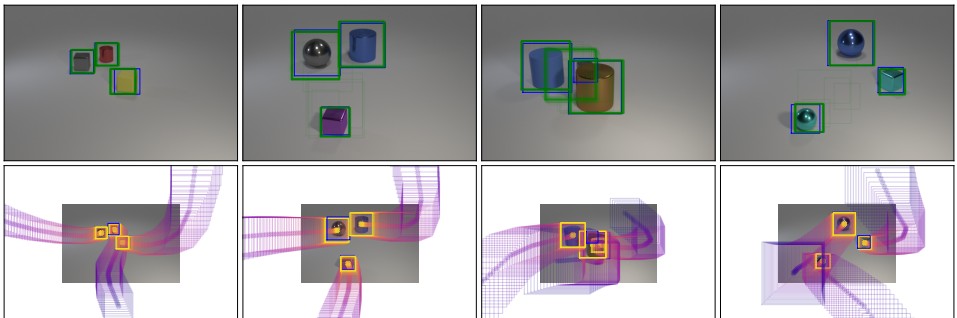

Figure 3: Bounding box prediction on CLEVR3 images. Each image shows 50 samples from our conditional flow (green, ground truth in blue), conditioned on the background image. The bottom row shows the trajectories of the boxes with time along the trajectory encoded by the color of the box.

terms: as the agents are steered onto the road, they have a higher density and thus a higher chance to collide. The opposite is equally true, as repelling agents can push each other off-road. As such it is not terribly surprising that the "single-only" ($g_\theta$-based) flow performs better when only considering offroad infractions. Nevertheless, the combined flow has the best performance overall, both in overall infraction rate and negative log likelihood.

To better understand the effects of our conditioning inputs, we report the performance of two ablations, one where the conditional input is only used in $g_\theta$ ("Cond. Single"), and one where it is only used in $f_\theta$ ("Cond. Pair"). In the former, the interactions between actors are independent of the scene, which one may expect to be a reasonable approximation. However, the results in Table 1 indicate that this input is in fact important, which may be understood from the fact that different locations lead to different traffic configurations. In the case where conditioning on $g_\theta$ is ignored, we also do worse than conditioning both, while surprisingly maintaining a fairly competitive result. Finally, we show a variant of our model where we condition the base distribution (i.e. $p_\theta(\mathbf{z}|\mathbf{y})$ vs. $p(\mathbf{z})$), but otherwise remove the conditional dependence from $f_\theta$ and $g_\theta$. Although in this case the collision rate is lowest, this can be attributed to the poor performance with respect to offroad infractions. Overall, this variation fails to provide a competitive result.

### 3.2.2 VARIABLE SET SIZE

Since our model is trained on a variety of different set sizes, and performs well for each of the different set sizes, we can investigate whether the it generalizes beyond the number of agents it has seen during training. We therefore generate samples in our roundabouts model with a previously unseen number of agents. Such samples are presented in the last column of Fig. 1, which have 28 agents, while the maximum number of agents present in the training data is 22. These results indicate the the inductive bias of the representation in Eq. (12) not only performs well in sample with respect to set size, but generalizes to larger set sizes too.

### 3.3 BOUNDING BOX PREDICTION

Our final experiment considers bounding box prediction. The gold standard in this sub-problem of object detection remains so-called "non-max-suppression," in which a large number of putative bounding boxes are scene-conditionally generated and then "pruned" by a greedy selection algorithm (Girshick et al., 2014; Ren et al., 2015). Some very recent studies (Hu et al., 2018; Zhang et al., 2019; Carion et al., 2020) have proposed the use of conditional set generators for object detection. We build on this idea by illustrating how our flow can be used to conditionally generate and score sets of object bounding boxes. Being able to compute the log density of a configuration of bounding boxes eliminates the need for a differentiable matching algorithm (Hu et al., 2018; Carion et al., 2020). Having a tractable density also opens up uncertainty-preserving approaches to downstream tasks such as outlier detection and object counting.

We assess the viability of our approach through the CLEVR (Johnson et al., 2017) dataset, which is a standard benchmark for set-generation models (Greff et al., 2019; Zhang et al., 2019; Locatello et al., 2020). While this former work combines localization with classification (i.e., predicting object

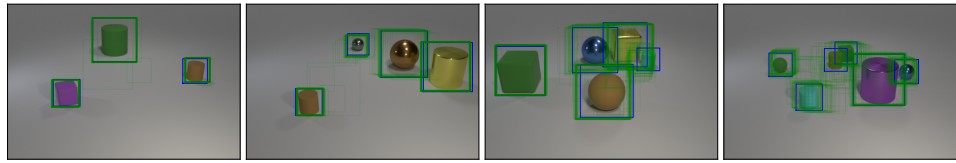

Figure 4: Bounding box prediction on CLEVR6 images. Each image shows 50 samples (green, ground truth blue) from our flow conditioned on the corresponding background image.

*position* and *type*), we focus on the task of bounding box prediction (i.e predicting object *position* and *size*). The CLEVR dataset does not provide bounding boxes, so we generate ground truth bounding boxes from object metadata. The full details on how to create bounding boxes for the CLEVR dataset are described in Appendix A.4.1.

We begin with a subset of the CLEVR dataset only containing three objects. We find that our flows perform well on this task, and show example predictions in Fig. 3. For each conditional image (displayed in the background), 50 samples from the conditional distribution are shown, graphically illustrating the variance of the conditional density. For each column, the bottom row displays the trajectory taken to generate a single one of these samples. The trajectories show the interactions between the bounding boxes over time, as they coordinate through the use of repulsive forces. Note that by sorting out which of the base distribution samples goes to which of the objects, the flow solves the assignment problem along the way. It is worth pointing out that the third sample has an occluded object, and the variance of the object position is clearly higher than that for objects where there is no occlusion, which we take to be evidence that these flows for bounding box prediction should be useful in uncertainty aware downstream applications. Importantly, the variance of the size is not significantly increased, which is correct behavior for this example. Additional samples are presented in the appendix. The averaged IOU with the ground truth is reported in Table 1b, and corresponds approximately to an average mismatch of 15% in each spatial dimension.

We continue our exploration with a larger subset of the CLEVR dataset, including images that have between three and six objects. This subset has also been used in Locatello et al. (2020) for object detection. We assume that the number of objects is given, and only predict the bounding box locations and sizes given the number of bounding boxes and the image. Some example samples are displayed in Fig. 4, with green boxes displaying samples from the distribution that is conditioned on the image in the background. We provide baseline results of Deep Set Prediction Networks (Zhang et al., 2019), which does not provide a tractable log density, in Appendix A.4.2. More samples are presented in Appendix A.3.5. The flow generalizes well over set cardinalities, hinting that some generalizing principles are learned by the flow about these bounding boxes interact, even with different set size. We moreover see that the more crowded the image becomes, the more spread there is in the predicted bounding boxes, representing increased uncertainty about object sizes and positions, also representing more occlusion. The overall IOU ("3" and "6"), as well as the IOU's separated by set cardinality ("6-3", etc.) are given in Table 1b. These results show that the flow trained on data with variable set size performs marginally better on the CLEVR3 subset than a flow trained only on that data, which we speculate is due to the larger amount of available data. A modest decrease in IOU can be observed as the sets become larger, resulting in an overall performance that is slightly lower.

## 4 DISCUSSION AND CONCLUSIONS

This work introduced conditional permutation invariant flows, a framework built on continuous normalizing flows that enables conditional set generation with a tractable density. We have applied our flows to two problems: realistic traffic scene generation, and bounding box prediction. For traffic scene generation, we significantly outperform baselines, and are the first to present a permutation invariant solution. Ablations to our flows highlight their intuitive mechanism, that can be understood as objects moving jointly in a field, augmented with pairwise interaction potentials. We have moreover shown that bounding box prediction can be enhanced with a tractable log density, opening an avenue to develop downstream vision algorithms that deal with uncertainty in a more principled way.

---

[1]By column from left to right, the flow samples are: top, top, top, bottom, both. For the last column in particular, our flow is able to produce more vehicles than ever appeared on that map in the training data.

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

## A  APPENDIX

### A.1  PROOF

We provide here the proof for Theorem 1.

*Proof.* We will consider equivariance of $T$ to a transposition of elements $i$ and $j$, denoted $\sigma_{i,j}$. A transposition $\sigma_{i,j}$ is a permutation for which $\sigma_i = j$, $\sigma_j = i$, and $\sigma_k = k$ for all $k \in \{1 \ldots N\} \backslash \{i, j\}$. Since any permutation can be constructed from a series of transpositions, proving that $T$ is equivariant to a transposition, trivially extends to equivariance to all permutations. In the following we have dropped the dependence on $\mathbf{y}$ and $t$ for notational clarity. We have

$$T(\sigma_{i,j}\mathbf{x}) = \sigma_{i,j}\mathbf{x} + \int_{t_0}^{t_1} \mathbf{v}_\theta(\sigma_{i,j}\mathbf{x})dt. \tag{14}$$

The first term $\sigma_{i,j}\mathbf{x}$ trivially satisfies the equivariance condition. Focusing on the $i^{\text{th}}$ term of the dynamics function $\mathbf{v}_{\theta,i}$

$$\mathbf{v}_{\theta,i}(\sigma_{i,j}\mathbf{x}) = \sum_{k\backslash\{i\}} f_\theta \left((\sigma_{i,j}\mathbf{x})_i, (\sigma_{i,j}\mathbf{x})_k\right) + g_\theta \left((\sigma_{i,j}\mathbf{x})_i\right) \tag{15}$$

$$= \sum_{k\backslash\{i,j\}} f_\theta \left((\sigma_{i,j}\mathbf{x})_i, (\sigma_{i,j}\mathbf{x})_k\right) + f_\theta \left((\sigma_{i,j}\mathbf{x})_{iv}, (\sigma_{i,j}\mathbf{x})_j\right) + g_\theta \left(\mathbf{x}_j\right) \tag{16}$$

$$= \sum_{k\backslash\{i,j\}} f_\theta \left(\mathbf{x}_j, \mathbf{x}_k\right) + f_\theta \left(\mathbf{x}_j, \mathbf{x}_i\right) + g_\theta \left(\mathbf{x}_j\right) \tag{17}$$

$$= \sum_{k\backslash\{j\}} f_\theta \left(\mathbf{x}_j, \mathbf{x}_k\right) + g_\theta \left(\mathbf{x}_j\right) \tag{18}$$

$$= \mathbf{v}_{\theta,j} \left(\mathbf{x}\right) = (\sigma_{i,j}\mathbf{v}_\theta(\mathbf{x}))_i, \tag{19}$$

thus demonstrating the dynamics are equivariant and

$$T(\sigma_{i,j}\mathbf{x}) = \sigma_{i,j}\mathbf{x} + \int_{t_0}^{t_1} \sigma_{i,j}\mathbf{v}_\theta(\mathbf{x})dt = \sigma_{i,j}T(\mathbf{x}). \tag{20}$$

For continuous normalizing flows, the inverse transformation $T^{-1}(\mathbf{x})$ is obtained by reversing the integration limits, and an identical derivation can be made to show $T^{-1}(\mathbf{x})$ is also equivariant. For the density, we have

$$\log p_{t_1}(\sigma_{i,j}\mathbf{x}(t_1)) = \log p_{t_0} \left(T^{-1}\left(\sigma_{i,j}\mathbf{x}(t_1)\right)\right) - \int_{t_0}^{t_1} \nabla_{\mathbf{x}} \cdot \mathbf{v}(\sigma_{i,j}\mathbf{x})dt. \tag{21}$$

Here, the divergence $\nabla_{\mathbf{x}} \cdot \mathbf{v}_\theta (\sigma_{i,j}\mathbf{x})$ denotes the divergence with respect to the argument of $\mathbf{v}$, evaluated at $\sigma_{i,j}\mathbf{x}$. Since the base distribution $p_{t_0}$ is permutation invariant, and $T^{-1}$ is equivariant

$$\log p_{t_0} \left(T^{-1}\left(\sigma_{i,j}\mathbf{x}(t_1)\right)\right) = \log p_{t_0} \left(\sigma_{i,j}T^{-1}\left(\mathbf{x}(t_1)\right)\right) \tag{22}$$

$$= \log p_{t_0} \left(T^{-1}\left(\mathbf{x}(t_1)\right)\right). \tag{23}$$

The divergence in the second term $\nabla_{\mathbf{x}} \cdot \mathbf{v}_\theta (\sigma_{i,j}\mathbf{x})$ is a sum over derivatives with respect to all its arguments, so is invariant to $\sigma_{i,j}$

$$\nabla_{\mathbf{x}} \cdot \mathbf{v}_\theta (\sigma_{i,j}\mathbf{x}) = \nabla_{\mathbf{x}} \cdot \mathbf{v}_\theta (\mathbf{x}). \tag{24}$$

We therefore have

$$\log p_{t_1}(\sigma_{i,j}\mathbf{x}(t_1)) = \log p_{t_1}(\mathbf{x}(t_1)), \tag{25}$$

thus showing that the dynamics are *equivariant*, and the density is *invariant*. $\square$

Table 2: Hyperparameters used for experiments. Abbreviations are defined in the appendix text.

| Experiment | $g_\theta$ | | $f_\theta$ | | $\mathbf{y}_{emb}$ | | | | |
|---|---|---|---|---|---|---|---|---|---|
| | $n$ | $h$ | $n$ | $h$ | $n$ | $c$ | $h$ | $t$ | batch |
| Example conditional | 5 | 200 | 5 | 200 | 3 | 16 | 200 | yes | 100 |
| Example bounding box | 5 | 200 | 5 | 200 | 3 | 16 | 200 | yes | 100 |
| Traffic baseline | 5 | 200 | 4 | 100 | 3 | 32 | 500 | no | 100 |
| CLEVR3 | 4 | 188 | 5 | 196 | 5 | 18 | 409 | no | 100 |
| CLEVR6 | 5 | 100 | 5 | 200 | 5 | 28 | 478 | no | 100 |

## A.2 EXPERIMENTAL DETAILS

### A.2.1 FORCE FUNCTION IMPLEMENTATION

We model the force functions $f_\theta(\mathbf{x}_i, \mathbf{x}_j)$ and $g_\theta(\mathbf{x}_i)$ by feed forward neural networks. For the pair force $f_\theta(\mathbf{x}_i, \mathbf{x}_j)$, we concatenate the inputs $\mathbf{x}_i$ and $\mathbf{x}_j$. In the case a time variable ($t$) is used, it is also concatenated. For the conditional input, we construct an embedding vector $\mathbf{y}_{\text{emb}}$, which we concatenate to the second layer inputs of $g_\theta$ and $f_\theta$. The embedding vector $\mathbf{y}_{\text{emb}}$ is generated using a separate neural network, here chosen to be a convolutional network, since all our conditional distributions have images as inputs.

### A.2.2 SOLVING THE ODE

We use the adaptive solver of Dormand and Prince of order 4 to solve the ODE (Dormand & Prince, 1980). To calculate the gradients of the ODE with respect to its parameters we use the adjoint method (Coddington & Levinson, 1955). This enables calculation of the gradients without back propagating through the computational graph. This functionality is all available in the `torchdiffeq` package (Chen et al., 2018), which is the implementation we use in our experiments.

### A.2.3 TABLE OF EXPERIMENTAL HYPERPARAMETERS

In all experiments $g_\theta$, $f_\theta$ are implemented as neural networks of $n$ layers and $h$ neurons per layer. The convolutional embedding network has $n$ layers of $c$ channels, followed by a single feed forward layer of $h$ neurons. We use sigmoid-linear units in all our dynamics functions, which satisfy the requirement of Lipschitz continuity, provided the networks are evaluated on a finite domain. The use of a time variable in the dynamics is indicated with $t \in \{\text{yes}, \text{no}\}$. For each experiment, these parameters are presented in Table 2.

### A.2.4 COMPUTATIONAL COST

Summing over all pairs of interactions is necessary to make the transformation permutation equivariant, but it comes at a quadratic cost in $N$. While not problematic for the set sizes in this study, this is clearly a limited approach for large numbers of objects. In these cases, it would be possible to set a boundary on the interaction range, or use a fixed set of $M < N$ inducing points, for a total cost of $MN$. Such approximations have been studied for example in transformers (which are also quadratic in the sequence length) (Vaswani et al., 2017; Wang et al., 2020). Furthermore, the divergences with respect to $\mathbf{x}_i$ are still quadratic with respect to $D$. This has been addressed in recent work by using functions that have divergences that can be easily evaluated using automatic differentiation (Chen & Duvenaud, 2019; Biloš & Günnemann, 2021). Although these types of functions are compatible with our framework, the current work only considers cases where $D \leq 5$, and therefore we do not implement it. Our overall algorithm therefore is of cost $N^2 D^2$. If it is necessary to construct distributions with larger $D$ (in for example object detection, rather than bounding box prediction), it is possible to use the methodologies from the aforementioned work to end up with a total computational cost of $MND$.

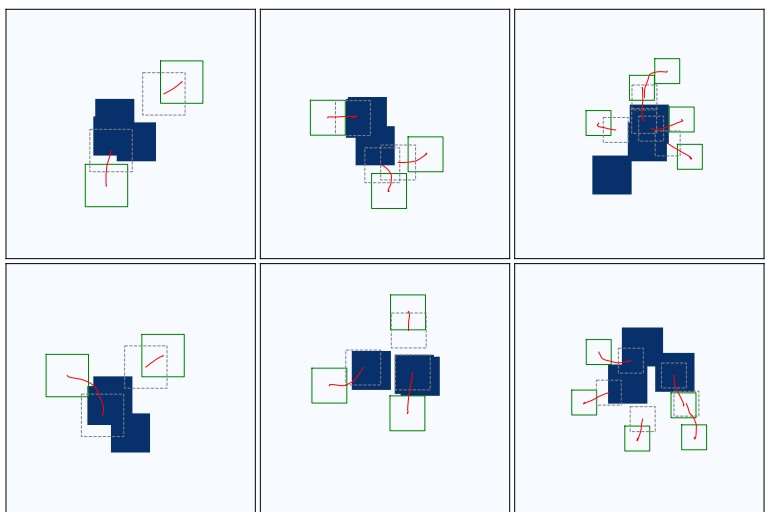

Figure 5: Conditional samples. The condition is an image, which is plotted as a blue on white background. The distribution is trained on samples that do not overlap with the blue regions, or with oneanother. The grey boxes are samples from the base distribution, the green boxes are samples from the flow. The red curves indicate the traveled trajectory for each box.

Table 3: Acceptance rates for conditional sampling. Results presented are the acceptance rate (AR) for Prior samples and the Conditional Permutation Invariant Flows (PIF)

| Set Size | Prior AR | PIF AR |
|:---:|:---:|:---:|
| 2 | 0.01 | 0.83 |
| 3 | $2.01 \cdot 10^{-3}$ | 0.79 |
| 5 | $1.76 \cdot 10^{-4}$ | 0.77 |

### A.2.5    COMPUTATIONAL RESOURCES

All our experiments were performed on a single GPU, all permutation invariant models were trained between 2 and 7 days of wall-clock time. The "vanilla" continuous normalizing flow, realNVP, and autoregressive model were trained over 14 days of wall-clock time.

### A.2.6    REJECTION SAMPLING

All samples presenting traffic scenarios in the main text of this manuscript have been checked by an automated procedure, and in a small amount of cases were rejected if an infraction occurred based on actors being offroad or vehicle overlap.

### A.2.7    DATASETS

The datasets used in this study are the INTERACTION datset (Zhan et al., 2019) (available for research purposes), and the CLEVR dataset (Johnson et al., 2017) (available under the Creative Commons CC BY 4.0 license).

### A.3    ADDITIONAL EXPERIMENTAL RESULTS

### A.3.1    CONDITIONAL NON-OVERLAPPING BOXES

Additional samples of of the conditional generation of non-overlapping boxes (Section 3.1) are presented in Fig. 5, for cardinalities 2, 3 and 5. Performance in terms of acceptance rate against an independent prior are reported in Table 3.

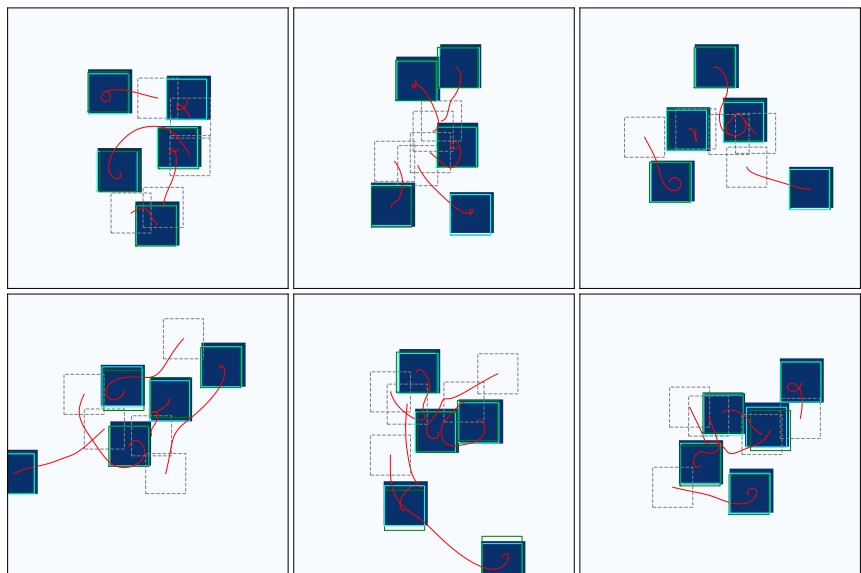

Figure 6: Additional samples of the example bounding box prediction problem.

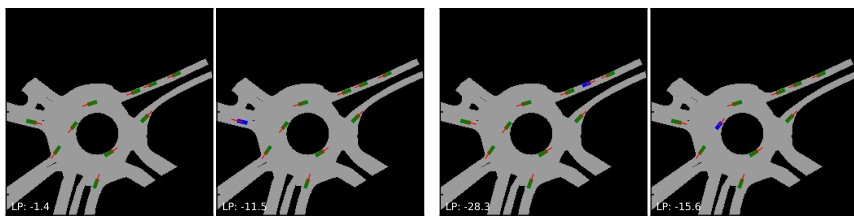

Figure 7: Sample and its log probability (left panel) and three corrupted variations where actors in blue have been turned around (last three panels).

### A.3.2 BOUNDING BOX PREDICTION

Additional samples for the example bounding box prediction task (Section 3.1) are presented in Fig. 6.

### A.3.3 OUTLIER DETECTION

Since our model has a tractable density, we can use it for outlier detection. In the traffic scene task, we study the case of mislabelled examples, which we artificially generate by rotating one of the actors in the scene by $\pi$. The original and corrupted scenes and corresponding log probabilities are shown in the last three panels of Fig. 7. It is clear that reversing one of the actors substantially decreases the probability. Moreover the model correctly captures the severity of the resulting infraction, which is less when an actor is going the wrong way on a two-way road, without the presence of other surrounding actors.

### A.3.4 REGULARIZATION

We present the effect of our regularization term on the bounding box prediction task, in which the effect is most pronounced. The proportionality constants of the $\ell^2$ and $\ell^2_{div}$ penalty terms are denoted as $\lambda$ and $\lambda_{div}$ respectively. Results for various $\lambda$ and $\lambda_{div}$ are presented in Fig. 8. The increase of the parameters $\lambda$ and $\lambda_{div}$ evidently creates more direct trajectories. We further find empirically that it drastically reduces the number of calls to the dynamics made by the ODE solver.

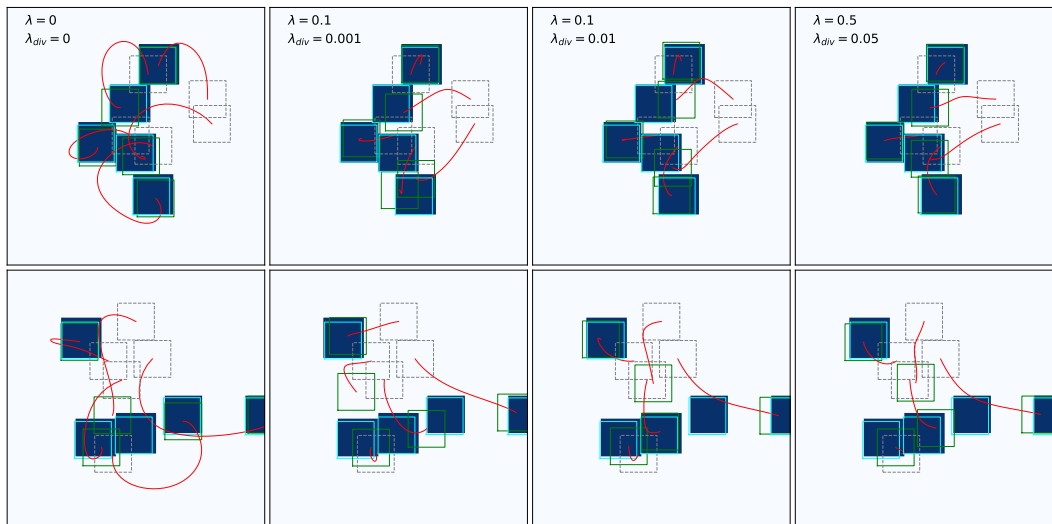

Figure 8: The effect of regularization on the dynamics. The penalty proportionality constants are reported per column in the top panel.

Table 4: Results for DSPN (Zhang et al., 2019)

| Set size | IOU |
|----------|------|
| 3 | 0.89 |
| 4 | 0.88 |
| 5 | 0.84 |
| 6 | 0.83 |

### A.3.5 CLEVR

Additional samples for the CLEVR3 dataset are presented in Fig. 9.

Additional samples for the CLEVR6 dataset are presented in Fig. 10.

### A.4 BOUNDING BOX PREDICTION

### A.4.1 CLEVR SIZE

The CLEVR dataset contains the pixel positions of objects, but not the bounding box sizes. The dataset does however provide the center locations of the objects in the global frame $\{x_i, y_i, z_i\}$. Since the objects are sitting on a flat plane at $z = 0$, its $z$ coordinate is equal to half its size. Furthermore, the dataset provides the distance to the camera along the viewpoint axis, $d_z$. Using these quantities, we approximate the bounding box size $\Delta$ as:

$$\Delta \approx \frac{z_i}{\sqrt{d_z}}. \tag{26}$$

We find empirically that this results in reasonable bounding boxes.

### A.4.2 CLEVR DSPN

We use Deep Set Prediction Networks (Zhang et al., 2019) to provide some experimental context for the CLEVR results. We note that this method does not provide a tractable log density, and therefore comparison between this method and ours is limited. The network was trained for 100 epochs and results are provided in Table 4 for set sizes reported in the main text.

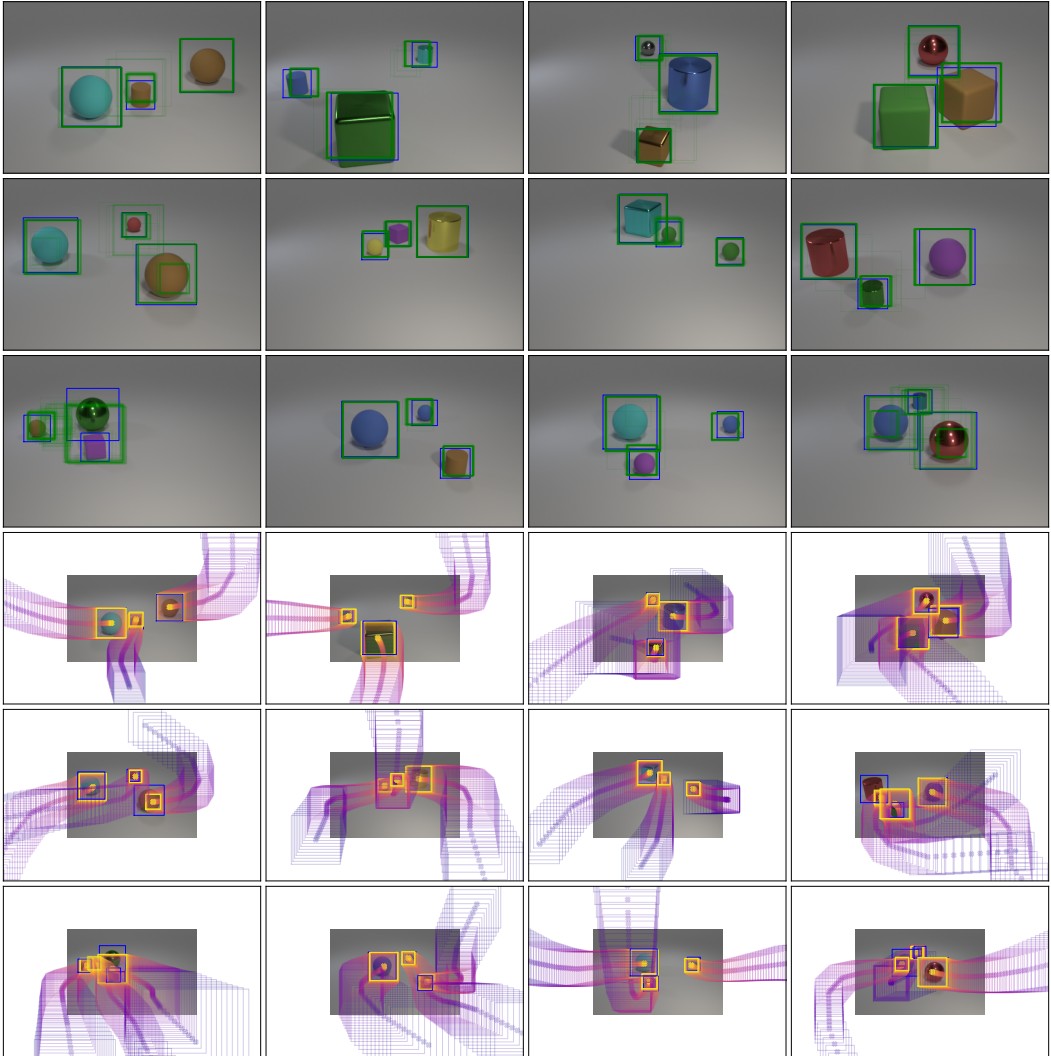

Figure 9: Additional examples for the CLEVR3 dataset. The blue boxes show ground truth bounding boxes, while the green boxes are all samples from the learned conditional distribution.

### A.4.3 DETECTION

In order to turn our bounding box detector in an object detection system, the objects need to be classified as well, which requires the use of categorical variables. While a full treatment of these is beyond the scope of this work, we provide some pointers to ways in which these could be included. Categorical variables are not trivially reparameterizable, but there has been a significant body of work that presents techniques to do this (Maddison et al., 2017; Hoogeboom et al., 2021), none of which provides an exact likelihood. Alternatively, the object classes could be treated as conditionally independent given the image and bounding box, in which case the likelihood is available, but this would neglect correlations between object classes other than those through the image and their bounding boxes.

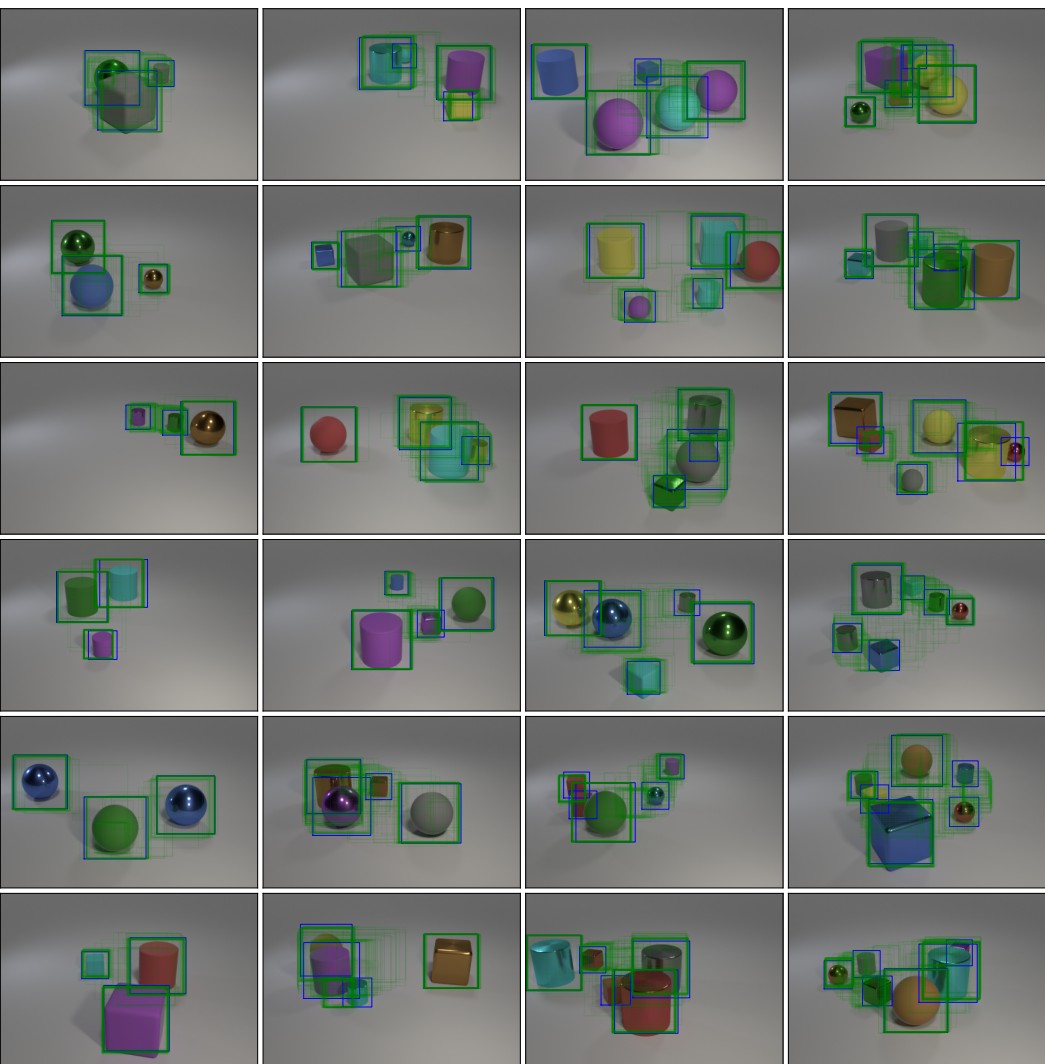

Figure 10: Additional examples for the CLEVR6 dataset. The blue boxes show ground truth bounding boxes, while the green boxes are all samples from the learned conditional distribution.

