# OpenReview forum: "Conditional Permutation Invariant Flows"
_ICLR.cc/2023/Conference — Submitted to ICLR 2023_

### Official Review · Reviewer_YVfE · 2022-10-18

**Confidence:** 4
**Correctness:** 4
**Technical Novelty And Significance:** 2
**Empirical Novelty And Significance:** 3
**Recommendation:** 5

**Clarity, Quality, Novelty And Reproducibility:**

Excellent clarity and quality. Novelty is questionable, as discussed above. Good reproducibility: the appendix provides extensive experimental details.

**Strength And Weaknesses:**

Strengths:
- The paper is of high quality and reads well. The introduction and background are particularly well done.
- The studied problem is important: modeling densities of sets is useful across a set of applications.
- Experiments on both pedagogical/synthetic and real datasets. Well-made figures and visualizations.
- Ablation experiments that study the role of the components of the proposed dynamics function.

Weaknesses:
- The novelty of the work is not clear to me. Prior work has considered both pairwise interactions and attention-like mechanisms (Kohler et al., 2020; Bilos et al., 2021) to build permutation invariant flows. What is the fundamental difference between the proposed method and prior work? If there is one, it is not made crystal clear in the paper.
- In line with the previous point, I'd expect the discussion of related methods to be deeper, and go beyond the claims that prior methods are evaluated on different objects (graphs), have additional invariances built in, or do not condition on side information (which is not difficult to do).
- Additional, non-generative baselines for the bounding box prediction problem would help put the results in a broader context.
- A single permutation invariant normalizing flow baseline is used for the traffic scenes problem: could additional baselines have been used?

Questions:
- Have you considered/experimented with standard neural network regularization methods (L1/L2, dropout, etc.) as an alternative to the regularization scheme proposed in section 2.3?
- Have you quantified the computational costs of the method? How do they compare to the baselines?

**Summary Of The Paper:**

The paper proposes a continuous normalizing flow architecture that is invariant to input permutations, which is useful for modelling densities of object _sets_. Authors design an equivariant dynamics function by only modelling point-wise and pair-wise interactions. Using this dynamics function with an invariant base distribution (like a standard normal) makes the modeled density permutation invariant. Authors evaluate the proposed flow on a set of synthetic and real tasks.

**Summary Of The Review:**

I (weakly) recommend rejecting the paper in its current form. While the manuscript is excellent, the novelty of the work is not clear to me. I would be happy to increase my rating if the authors clarified this, and updated the text accordingly.

---

> ### Author Response · Authors · 2022-11-15
> **Response to YVfe**
>
> We thank reviewer YVfe for their time and attention and will respond to their concerns point by point.
>
> **Weaknesses**
>
> >The novelty of the work is not clear to me. Prior work has considered both pairwise interactions and attention-like mechanisms (Kohler et al., 2020; Bilos et al., 2021) to build permutation invariant flows. What is the fundamental difference between the proposed method and prior work? If there is one, it is not made crystal clear in the paper.
> In line with the previous point, I'd expect the discussion of related methods to be deeper, and go beyond the claims that prior methods are evaluated on different objects (graphs), have additional invariances built in, or do not condition on side information (which is not difficult to do).
>
> Regarding the novelty of our work, we point the reviewer to our general response.
>
> >Additional, non-generative baselines for the bounding box prediction problem would help put the results in a broader context.
>
> We have an additional baseline in the appendix of our work, but regretfully forgot to mention this in the main text, this has now been corrected. We stress here also our response to the other reviewers: the point here is not actually the generative nature of our model, but rather the ability to compute the log-density. As our baseline model does not provide a density, we have put comparison to it in the appendix. We are unaware of any other work that does bounding box prediction with tractable density, so no further baselines are provided.
>
> > A single permutation invariant normalizing flow baseline is used for the traffic scenes problem: could additional baselines have been used?
>
> First, we present at least two permutation invariant normalizing flows (technically the Gaussian distribution is permutation invariant too): the ablations of our work constitute flows that represent conditional versions of the work of Li et al (2021) and Bilos (2021). We are unaware of any other permutation invariant flows that fit our problem structure. If the reviewer can suggest others, we may consider implementing and adding these results if time allows.
>
> **Questions**
> >Have you considered/experimented with standard neural network regularization methods (L1/L2, dropout, etc.) as an alternative to the regularization scheme proposed in section 2.3?
>
> We have not considered these forms of regularization in our work for the simple reason that the goal of our regularization is not to prevent overfitting (which dropout, and L1/L2 are focussed on), but rather to take the shortest path possible between a latent sample z and a variable x to make the integration cheaper to evaluate. The penalty term we apply directly targets the path length, by minimizing its length (l2), and the total deformation of the space (l2div), without incurring significant overhead. Nevertheless, we think L1/L2 and dropout are compatible with our method and may be another technique in the user's toolbox, but leave its evaluation for future work.
>
> >Have you quantified the computational costs of the method? How do they compare to the baselines?
>
> While we have not performed an exhaustive analysis of the computational costs of our models versus baselines, we find similar evaluation times for all variations and ablations of our permutation invariant flow and the autoregressive model. The Gaussian baseline is up to 4 times faster to evaluate, and the non-permutation invariant continuous flow, and the RealNVP baseline are both approximately a factor 5 slower. All these estimates are for a fixed set cardinality of 7.

---

### Official Review · Reviewer_VEu3 · 2022-10-24

**Confidence:** 4
**Correctness:** 3
**Technical Novelty And Significance:** 2
**Empirical Novelty And Significance:** Not applicable
**Recommendation:** 5

**Clarity, Quality, Novelty And Reproducibility:**

I have no concerns for the correctness, clarity and quality of the paper. The technical novelty of the paper is very limited. Please refer to strength and weakness for more details.

**Strength And Weaknesses:**

Strengths:

1. The paper is well organized and well written.
2. Experiment results show strong performance of the model.


Weakness:

1. The technical novelty of the paper is very limited. Graph normalizing flows and its continuous versions have been studied[1, 2]. The permutation equivariance/invaraince property of normalizing flows has also been studied before [3]. Clearly, the work is just an interesting application of existing models to set data. Moreover, the transformation of the continuous normalizing flow being used in the work is actually equivariant instead of invariant. What is invariant is the density. The title and the naming of the model are misleading. The author should be more precise about their contributions.
2. Bounding box prediction results given the image are deterministic; it is not a conditional generative task. I’m not fully convinced this is a good task to study a conditional normalizing flow model.

[1] Deng, Zhiwei, et al. "Continuous graph flow." arXiv preprint arXiv:1908.02436 (2019).

[2] Xhonneux, Louis-Pascal, Meng Qu, and Jian Tang. "Continuous graph neural networks." International Conference on Machine Learning. PMLR, 2020.

[3] Biloš, Marin, and Stephan Günnemann. "Scalable normalizing flows for permutation invariant densities." International Conference on Machine Learning. PMLR, 2021.

**Summary Of The Paper:**

The paper proposes a permutation invariant conditional normalizing flows based on a continuous version of graph normalizing flows. The proposed model is evaluated on the tasks of generating realistic traffic scenes and bounding box prediction.

**Summary Of The Review:**

The paper is well written and experiment results demonstrate the proposed conditional permutation invariance normalizing flows. However, the technical novelty of the proposed work is limited and the claims of contributions should be more precise.

---

> ### Author Response · Authors · 2022-11-15
> **Response to VEu3**
>
> We thank the reviewer for their time and attention and will respond to each of their concerns individually.
>
> >The technical novelty of the paper is very limited. Graph normalizing flows and its continuous versions have been studied[1, 2]. The permutation equivariance/invaraince property of normalizing flows has also been studied before [3]. Clearly, the work is just an interesting application of existing models to set data. Moreover, the transformation of the continuous normalizing flow being used in the work is actually equivariant instead of invariant. What is invariant is the density. The title and the naming of the model are misleading. The author should be more precise about their contributions.
>
> We thank the reviewer for bringing [1] and [2] to our attention. Regarding [1], this preprint is aimed at graphs, and it is unclear how the connectivity matrix is generated. Assuming the connectivity matrix is taken from ground truth, with a dense connectivity matrix it constitutes an unconditional version of our work that only contains the pair interaction terms (f(x1,x2)), but no global term (g(x)), which we show performs worse as one of the ablations in Table 1. However, since it is not completely clear how the connectivity matrix is generated, we will omit this from related work for now, but invite the reviewer to expand on their opinion regarding the connection. Regarding [2], this is interesting work but regards classification of graph nodes, which is a different problem than the generative and density estimation one we consider in this work. We compare to [3] in our work already, and provide a baseline that shows its performance is worse than ours.
>
> Regarding overall novelty: please see our general response regarding novelty.
>
> Regarding the title of the paper, we use the term "normalizing flow" in an overloaded sense, such that it means both the transformation, and the distribution and density. The reviewer correctly points out that the first one should be equivariant, and the latter two should be called invariant. Our title refers to normalizing flows in the distribution object sense of the word, hence our title reads "invariant", rather than "equivariant"
>
> >Bounding box prediction results given the image are deterministic; it is not a conditional generative task. I’m not fully convinced this is a good task to study a conditional normalizing flow model.
>
> We would like to point out that while our flow can be used in a generative sense, its strength in this task is the ability to compute the log density. This means we can use our flow to generate samples from the inferred distribution, but also use this distribution for downstream tasks that require knowledge of uncertainty. Arguably, there is limited uncertainty in our object localization task, but we chose this specific task because other work has used this as a set-generation test bed in the past, so there is precedence from existing literature. As we alluded to in our response to g69S, we think that our method would really shine in a 3D detection task, where uncertainty is an inherent part of the problem. However, such a task should contribute a publication in itself and we leave its investigation for future work.

---

### Official Review · Reviewer_g69S · 2022-10-25

**Confidence:** 3
**Correctness:** 3
**Technical Novelty And Significance:** 2
**Empirical Novelty And Significance:** 3
**Recommendation:** 6

**Clarity, Quality, Novelty And Reproducibility:**

Very clear paper.  The method appears to be novel based on comparison to related work (I may not be aware of other related work.)  While most of the elements have appeared before (or very similar) the combination is new.  No code is provided, although the method and experiments are clearly described and most experiments are performed on publicly available data.

**Strength And Weaknesses:**

## Strengths
- The proposed method is simple and well-explained.  The need for permutation-invariance is well motived in the examples (allows generalization to different numbers of objects, avoids matching loss) and mathematically reasonable as inductive bias (it avoid makes arbitrary ordering decisions).  The addition of conditioning is also well motivated in the examples.  For example, in the traffic generation case, it is clear that map information will be relevant to the car location distribution.
- The traffic scene experiment is pretty thorough including several non-equivariant baselines.  The results are convincing that permutation-equivariance is useful for the task.  Moreover, the results show good generalization over the number of cars.  The ablation shows the usefulness of the different global force term and pairwise interaction term and the analysis is insightful in noting the utility of each for staying on the road and not colliding.  I think the domain-relevant metrics are useful here.


## Weaknesses
- The effect of the efficient divergence computation in 2.2 and regularization in 2.3 are not evaluated in the experiments.
- It would nice to have a permutation-equivariant baseline for the traffic scene generation task.  For example a non-flow based method.
- It's not clear if the non-equivariant baselines in traffic scene generation are trained with ample data augmentation.
- The bounding box prediction experiment has some potential issues.  There are no baseline comparisons here.  Only IOU is presented and with no variance.  Also, it is a bit of a strange task given that the target distribution is a point and so the fit distribution only represents uncertainty.  However, no serious evaluation of how well the model capture uncertainty (calibration, e.g.) is done.  On the plus side, this experiment does show the method can work with real world data.
- It would be good to have some quantitative / qualitative evaluation on sample diversity and coverage.

## Questions
- The velocity term is simple, but seems quite constrained.  Are there potential issues with expressivity given this simple form imposed by equivariance and the fact the flow dynamics give a diffeomorphism of the space?  Even if theoretically expressive, could the flow dynamics have trouble fitting certain kinds of distributions from certain priors?
- Can you say more the about the motivation to generate traffic scenes?   It's not completely clear to me how it is useful.
- Can this method be applied to high-dimensional modeling with many objects or high-dimensional object features?  For example, could it be used for trajectory prediction with 100 objects?
- Although this method is not specialized for it, could it be used for and compared to methods that generate point clouds or graphs?
- For table 1 (b), I didn't fully understand the Data column.  What does "6-4" mean?

## Minor Points
- I would write $v_{\theta,i}(x)$ on the line after equation (9).
- Caption in Fig. 2 described the right two panels as "to avoid the". Should it be "to bound the"?
- Page 6, Para 2, Line 5, typo "flow flow"
- I liked the guessing game in figure 1.


**Summary Of The Paper:**

This work proposes a continuous normalizing flow model, an exact likelihood generative modeling method, which is conditional and permutation invariant.  The flow is driven by dynamics computed from a shared global force term and a pairwise interaction term which have shared weights across all elements thus achieving permutation equivariance in a similar manner to DeepSets or a message-passing NN over a fully connected graph.    The model is tested over two toy datasets involving box placement and more realistic datasets involving traffic scene generation and bounding box placement.  The dataset outperforms non-permutation invariant baselines in the traffic scene generation experiment in both NLL and domain relevant metrics.

**Summary Of The Review:**

I felt the method was well-described, straightforward, and wholly suited to the a range of tasks where conditioning and permutation-invariance are useful properties.  The experimental results support the value of incorporating these features.  Overall, the weaknesses are largely minor and addressable.  (I'd probably opt for 7 if available, so I'll be conservative for now, but would be happy to upgrade my score if no serious concerns are raised.)

---

> ### Author Response · Authors · 2022-11-15
> **Response to g69S (2 / 2)**
>
> **Questions**
> >The velocity term is simple, but seems quite constrained. Are there potential issues with expressivity given this simple form imposed by equivariance and the fact the flow dynamics give a diffeomorphism of the space? Even if theoretically expressive, could the flow dynamics have trouble fitting certain kinds of distributions from certain priors?
>
> We speculate in our introduction that this could indeed be the case: equivariance does indeed limit the functional forms that can be used, and it is true there exists a functional form of the dynamics v that encompasses our formulation in terms of re-used f and g. However, we also point out in our introduction (and our experiments confirm this), that such a function would be impractical, to the point of being unusable. Memory constraints aside, a flow of that type would likely need to visit all permutations of data, resulting in a factorially increased training time. Regarding the issues that arise with diffeomorphisms, we agree that it is possible we lose some performance (see also our answer regarding the object localization task, we have clarified a baseline), but we think the tractable density we get in return is worth that sacrifice.
>
> >Can you say more the about the motivation to generate traffic scenes? It's not completely clear to me how it is useful.
>
> Firstly, please refer to the recent paper of Tan (2021), which considers the same problem, but instead decides on an autoregressive model (which we in fact compare to in our baselines). Secondly, there is a host of work that does autoregressive modelling across time of traffic scenarios. Playing out such scenarios (e.g to train reinforcement learning models) relies on generating a valid starting point, which an autoregressive model (p(x_t+1 | x_t)) requires. Sampling such elements from the dataset is theoretically possible, but limits the number of scenes to the dataset size, and can not go beyond maps and/or agent number seen in the dataset. Hence, our model is extremely useful in developing algorithms with autonomous driving applications.
>
> >Can this method be applied to high-dimensional modeling with many objects or high-dimensional object features? For example, could it be used for trajectory prediction with 100 objects?
>
> For expositional clarity let us briefly repeat that in our notation the dimensionality of the problem is NxD, with N the number of objects in the set, and D the dimensionality of each object. While our approach can be used in theory for higher dimensional objects (i.e. large D), our formulation is particularly well suited for problems with relatively low dimension (low D) of each set object, and small to medium sized (i.e. N) strongly interacting sets. While our approach satisfies two of the requirements for trajectory modelling (medium N, strongly interacting), it is not particularly suited for predicting full time series in one shot (large D). We think however that this is a great suggestion, and we think that a combination of the techniques developed in Bilos et al (which focus on forms of the dynamics for which the jacobian is cheap to evaluate with respect to D) and our form for the dynamics (which is cheap to evaluate with respect to N, yet allows meaningful interactions) would be an excellent solution. We will leave a full exploration of such a scenario for future work.
>
> >Although this method is not specialized for it, could it be used for and compared to methods that generate point clouds or graphs?
>
> Following on from our previous answer, while our technique can theoretically be used for point cloud generation, it is again not where our method shines. Point clouds generally are large collections of elements (i.e. large N), in which the interaction between objects themselves is not very strong. As a matter of fact one piece of prior art "Pointflow" (Li, 2021) that focuses on point clouds can be seen as an ablation of our work, that ignores the pair interaction term f altogether, which results in a linear cost in N, but forgoes any interactions between elements themselves.
>
> >For table 1 (b), I didn't fully understand the Data column. What does "6-4" mean?
>
> Our model is trained on a collection of different set cardinalities simulataneously (in this case, 3- 6), and we can run the evaluation separately for each of these cardinalities, i.e. 6-4 refers to the evaluation on cardinality 4, while the model was trained to predict 3, 4, 5, and 6. We thank the reviewer for pointing out this was unclear, we have added some text that we hope clarifies this.
>
> **Minor**
> >I would write  on the line after equation (9).
> >Caption in Fig. 2 described the right two panels as "to avoid the". Should it be "to bound the"?
> >Page 6, Para 2, Line 5, typo "flow flow"
> >I liked the guessing game in figure 1.
>
> Thank you for pointing out these minor issues and your careful reading. We have addressed them in our updated manuscript. We are glad you hear you enjoyed the guessing game in Fig. 1
> (2/2)

---

> > ### Comment · Reviewer_g69S · 2022-12-06
> > **Thanks for the response**
> >
> > Thank you very much for this thorough response which has helped clear up some points for me including evaluations of 2.2 and 2.3, motivation, and limitations of the method.  While I think the experiments could be stronger, it continues to seem to me the work contains a novel combination of attributes (permutation equivariant, conditional, flow-based) which could be applicable and useful for other authors, and so I favor acceptance.

---

> ### Author Response · Authors · 2022-11-15
> **Response to g69S (1 / 2)**
>
> We thank reviewer g69S for their feedback and comments, and will respond point by point.
>
> **Weaknesses**
> >The effect of the efficient divergence computation in 2.2 and regularization in 2.3 are not evaluated in the experiments.
>
> We point the reviewer to results in appendix A.3.4, where the effect of the regularization penalty in section 2.3 is (qualitatively) evaluated. The construction of dynamics with tractable divergences to mitigate the noise of Hutchinson's trace estimator has been the topic of previous work (Bilos 2021, Chen 2019) so we decided to not repeat that here. Nevertheless, we agree that adding explicit experiments that show the effect of exact divergence would be desirable and we will attempt to add these to our updated manuscript if time allows.
>
> >It would nice to have a permutation-equivariant baseline for the traffic scene generation task. For example a non-flow based method.
>
> We are unaware of any other models that are conditional, equivariant, generative models with a tractable log-density. As a matter of fact, this is the premise our novelty claim is based on. We are open to running such experiments if the reviewer can suggest any methods that would be suitable.
>
> >It's not clear if the non-equivariant baselines in traffic scene generation are trained with ample data augmentation.
>
> For our autoregressive model, the data is first sorted along the first positional dimension, in line with the strategy adopted in Tan et al. (2021). In the case of our RealNVP, non-permutation invariant flow, and Gaussian baselines, the set elements are randomly permuted, meaning they appear differently every epoch. The baselines are trained until no more improvement of the validation signal is observed. The total amount of training steps is similar in all cases. The non-equivariant methods have certainly not observed all random permutations of each datapoint, as such extensive data augmentation would imply factorially many training steps, which is simply not possible to do. Moreover, since the validation signal stopped improving, we highly doubt (and allude to in our introduction) such data augmentation would be effective.
>
> >The bounding box prediction experiment has some potential issues. There are no baseline comparisons here. Only IOU is presented and with no variance. Also, it is a bit of a strange task given that the target distribution is a point and so the fit distribution only represents uncertainty. However, no serious evaluation of how well the model capture uncertainty (calibration, e.g.) is done. On the plus side, this experiment does show the method can work with real world data.
>
> We agree with the reviewer that there is room for improvement in our localization task. Since scene generation is a more obvious choice for an uncertainty based model, we focussed most of our efforts and baselines there. The reasons for including this experiment are 1. it has been used as a test-bed in previous work developing models with set-valued outputs (Greff 2019, Locatello 2020, Zhang 2019) and 2. it is clear that successful application of our technique in for example 3d detection would be a huge benefit to deal with uncertainty in a principled way. We think such an investigation is worthy of a publication in itself and therefore will leave this for a future submission.
>
> >It would be good to have some quantitative / qualitative evaluation on sample diversity and coverage.
>
> We can potentially add more samples of the initial conditions scene to the appendix, but we are unaware of any quantitative methods to evaluate sample diversity on this specific task. Regarding coverage, we would like to point the reviewer to the rightmost panel in Fig. 1, where both samples are generated with a previously unseen number of agents, showcasing the ability to extend beyond data seen during training.
>
> (1 / 2)

---

### Author Response · Authors · 2022-11-15
**General response**

**General response and a comment regarding novelty**

We first of all thank the reviewers for their time and attention. It appears that all reviewers agree the paper is well written and communicates material clearly, which we are very happy to hear. It seems there are no technical issues, and generally the work is considered to be of high 	quality. However, there appears to be concern about the novelty of our work:  it seems that two of the reviewers think the work is incremental, or otherwise the novelty is not clearly communicated.

We would like to address this point of our work generally, as it appears to be a point of concern for two out of three reviewers. Let us first repeat the statement in the last two sentences of the bottom paragraph on page 1: the novelty of our work is to create a conditional normalizing flow over set valued data. It is indeed true that various elements (but not all) of our flow have appeared in previous work, as is clearly pointed out in our related work section.

Two reviewers argue that using inspiration from existing work and making these mechanisms depend on external input so that they can be used in two industry-ready applications is insufficient to warrant a publication in itself. We respectfully disagree, for the following reasons:

1. These unconditional models by themselves are completely unsuitable for the tasks we consider here: it goes without saying that unconditional scene generation model or an unconditional bounding box generation model is utterly useless. Hence, conditioning is the element of this work that makes it applicable to practical applications, of which we provide two compelling demonstrations.
2. There are ample examples of high impact, prior art that are based on the premise of conditioning existing generative models on external inputs. Including ones where neither the conditioning mechanism nor the generative model were new by themselves, but rather the combination of them was the novelty. A non-exhaustive list would include text to image generation [1,2], conditional image generation [3], trajectory modelling [4], and even traffic scene generation [5]. The amount work in this category is vast, and certainty too large to summarize to any extent in this reply, but the above serve as some examples of the type.

In light of these previous publications, the fact that the related work we build on is completely useless without such conditioning, and the fact that at least one of the reviewers appears to agree with us, we hope the other reviewers will reconsider their opinions about the value of such papers, as we think they form an interesting important, and particularly useful body of work.

[1] Ramesh, Aditya, et al. "Zero-shot text-to-image generation." International Conference on Machine Learning. PMLR, 2021.

[2] Mansimov, Elman, et al. "Generating images from captions with attention." arXiv preprint arXiv:1511.02793 (2015).

[3] Harvey, William, Saeid Naderiparizi, and Frank Wood. "Conditional Image Generation by Conditioning Variational Auto-Encoders." International Conference on Learning Representations. 2021.

[4] Bertugli, Alessia, et al. "AC-VRNN: Attentive Conditional-VRNN for multi-future trajectory prediction." Computer Vision and Image Understanding 210 (2021): 103245.

[5] Tan, Shuhan, et al. "Scenegen: Learning to generate realistic traffic scenes." Proceedings of the IEEE/CVF Conference on Computer Vision and Pattern Recognition. 2021.

---

### Decision · Program_Chairs · 2023-01-20

**Decision:**

Reject

**Justification For Why Not Higher Score:**

N/A

**Justification For Why Not Lower Score:**

N/A

**Metareview: Summary, Strengths And Weaknesses:**

Thank you for submitting your work to ICLR 2023.
After a discussion among the reviewers we found this work not ready for publication.
Although the reviewers appreciated the simplicity and elegance of the method, the presented experiments, and found the conditioning part interesting, they still had some lingering concerns: the model construction and training, that takes the major part of the paper is incremental, while the conditioning part is short and straightforward. We would recommend the authors to restructure the paper, expanding the conditioning part, and adding further "real-life" applications and baselines.

**Summary Of Ac-Reviewer Meeting:**

The reviewers found the paper incremental where the most interesting point (conditioning of the model) is rather small in space. They want a rather major revision where the paper is restructured, further expanding the conditioning part, and applications/baselines added.